# TCMacro: A Simple and Robust ImageJ-Based Method for Automated Measurement of Tail Coiling Activity in Zebrafish

**DOI:** 10.3390/biom11081133

**Published:** 2021-08-01

**Authors:** Kevin Adi Kurnia, Fiorency Santoso, Bonifasius Putera Sampurna, Gilbert Audira, Jong-Chin Huang, Kelvin H.-C. Chen, Chung-Der Hsiao

**Affiliations:** 1Department of Bioscience Technology, Chung Yuan Christian University, Chung-Li 320314, Taiwan; kevinadik-adi@hotmail.com (K.A.K.); fiorency_santoso@yahoo.co.id (F.S.); boni_bt123@hotmail.com (B.P.S.); gilbertaudira@yahoo.com (G.A.); 2Master Program in Nanotechnology, Chung Yuan Christian University, Chung-Li 320314, Taiwan; 3Department of Chemistry, Chung Yuan Christian University, Chung-Li 320314, Taiwan; 4Department of Applied Chemistry, National Pingtung University, Pingtung 900391, Taiwan; hjc@mail.nptu.edu.tw; 5Center for Nanotechnology, Chung Yuan Christian University, Chung-Li 320314, Taiwan; 6Research Center for Aquatic Toxicology and Pharmacology, Chung Yuan Christian University, Chung-Li 320314, Taiwan

**Keywords:** ImageJ, chorion, tail coiling, ROI, zebrafish, embryo

## Abstract

Tail coiling is a reflection response in fish embryos that can be used as a model for neurotoxic analysis. The previous method to analyze fish tail coiling is largely based on third-party software. In this study, we aim to develop a simple and cost-effective method called TCMacro by using ImageJ macro to reduce the operational complexity. The basic principle of the current method is based on the dynamic change of pixel intensity in the region of interest (ROI). When the fish tail is moving, the average intensity is increasing. In time when the fish freeze, the peak of mean intensity is maintaining at a relatively low level. By using the optimized macro settings and excel VBA scripts, all the tail coiling measurement processes can be archived with few operation steps with high precision. Three major endpoints of tail coiling counts, tail coiling duration and tail coiling intervals can be obtained in batch. To validate this established method, we tested the potential neurotoxic activity of Tricaine (methanesulfonate, MS-222) and psychoactive compound of caffeine. Zebrafish embryos after Tricaine exposure displayed significantly less tail coiling activity in a dose-dependent manner, and were comparable to manual counting through the Wilcoxon test and Pearson correlation double validation. Zebrafish embryos after caffeine exposure displayed significantly high tail coiling activity. In conclusion, the TCMacro method presented in this study provides a simple and robust method that is able to measure the relative tail coiling activities in zebrafish embryos in a high-throughput manner.

## 1. Introduction

Zebrafish is known for its versatility as an animal model due to its capability of being used on locomotor activity, physiological and developmental research that can be used for drug screening. During its lifetime, tail coiling (TC) is the first locomotor activity to be observed in zebrafish that originated from a newly formed singular neural circuit during the developmental process. Tail coiling can be observed starting from 17 h post-fertilization (hpf) at the pharyngula stage, however, TC activity does not react to light or touch stimuli until 21 hpf [1]. The use of zebrafish embryos in research also has several benefits, such as minimizing exposure solution quantity, fast developmental process and body transparency, which make it becomes an excellent animal model for developmental, pharmacological and toxicological studies [2,3,4]. Zebrafish embryos have also been proven able to detect neurotoxic effect via TC measurement [5,6,7]. Furthermore, data collected on zebrafish embryos can also be extrapolated to humans, reinforcing its’ capability as an animal model [5]. Furthermore, a previous study by Selderslaghs et al. verified the predictive power of TC, as it is able to show comparable locomotor activity stimulation results compared to zebrafish larvae (5–6 dpf) [8]. However, the timing of chemical introduction to zebrafish embryos should also be taken into account as it might affect zebrafish embryo development [9].

Currently, there are numerous available methods to obtain TC data in zebrafish. These methods mainly involve video recording using a charge-coupled device (CCD) camera and counting TC incidents using either manual [10,11,12,13] or machine-assisted methods [6,7,14,15,16,17]. Currently, there are several available methods for observing TC activity based on either third-party or open-access software (Table 1). One of these open-access methods proposed by González-Fraga et al. uses MATLAB as a major platform which requires knowledge on coding [14]. The other method, called EMAnalysis proposed by Zhang et al. used ImageJ, a common platform for image analysis [16]. However, in their publication, an internet connection was still required for data processing. Therefore, we want to provide a new way for TC data collection using ImageJ (FIJI) which will be available offline and able to obtain comparable results to the manual methods. This proposed method would use dynamic pixel changes to obtain the TC activity. Through the optimized ImageJ segmentation process, color adjustment and filtering, we were able to obtain a suitable image for optimal detection and automatic region-of-interest (ROI) selection results. Subtracted image stack would then be produced from the video to obtain TC data. Afterward, the data will be saved to set the destination in .xlsx form for final processing in Microsoft Excel 2016. In order to validate our proposed method, we compared the results obtained using the TCMacro method to the manual counting method which is one of the most common methods to collect TC data [10,11,12,13]. MS-222 (Tricaine methanesulfonate) and caffeine was used to validate our method as it is known to be a compound capable of reducing and increasing locomotor activity, respectively [14,15].

## 2. Materials and Methods

### 2.1. Zebrafish Maintenance

Wild-type zebrafish were used in this research. Zebrafish were maintained in a continuously filtered and aerated water system with a 10/14 of dark/light cycle and water temperature at ~26 °C. Zebrafish eggs were obtained by breeding male and female zebrafish at a 2:1 ratio. Zebrafishes used for breeding were separated using a separator to control the breeding time. Harvested eggs were kept in an incubator at 28 ± 1 °C until ~6 h post-fertilization (hpf). After ~6 h, living zebrafish eggs were selected to be exposed to the chemicals.

### 2.2. Chemical Preparation and Exposure

Tricaine Methanosulfonate (MS-222) (Shanghai Aladdin Bio-Chem Technology Co., Ltd., Shanghai, China) was prepared by dissolving in ddH_2_O to make 1 mg/mL stock solution and stored at 4 °C until use. MS-222 stock solution was diluted to 1 × 10^−7^ mg/mL (0.1 ppb), 1 × 10^−6^ mg/mL (1 ppb) and 1 × 10^−3^ mg/mL (1 ppm) as zebrafish locomotor activity inhibitor. Previous study used 0.01 mg/mL, 0.05 mg/mL and 0.1 mg/mL MS-222, thus, the dose we used was 100× lower [15]. MS-222 is an anesthesia and euthanasia for zebrafish embryos and larvae and it works by blocking gill ventilation on adult zebrafish. Zebrafish embryos and larvae can use the cutaneous gas exchange to fulfill their oxygen demand during MS-222 exposure [18]. To slow down tail coiling, zebrafish embryos aged at ~6 hpf were exposed to 10 mL of each MS-222 concentration within a clean petri dish. The dish was covered with a lid to prevent cross-contamination and kept in an incubator at 28 ± 1 °C until 22–24 hpf. To speed up tail coiling activity, caffeine (Sigma-ALDRICH, St. Louis, MO, USA) was prepared by dissolving in ddH_2_O to make a 1 mg/mL stock solution. The caffeine stock solution was further diluted to working solution at 0.15 mg/mL (150 ppm), 0.3 mg/mL (300 ppm) and 0.6 mg/mL (600 ppm) and applied to zebrafish embryos as the method conducted for MS-222.

### 2.3. Agarose Wells Preparation

Agarose wells (1% concentration) were made using a 3D-printed custom mold for microinjection purpose. Agarose gel was prepared in a petri dish by adding 0.2 g of agarose powder with 20 mL of ddH_2_O and microwaving the mixture for 3 min until the agarose powder dissolved. Afterward, the mixture was poured into a petri dish and the custom mold was placed immediately. After the agarose gel had solidified, the mold was taken off, the gel was covered with water and the petri dish lid was closed to prevent cross-contamination and kept the gel from drying. Lastly, the agarose gel was stored in the refrigerator at ~4 °C until further use.

### 2.4. Video Recording and Conversion

Two recording setups were used in this experiment. The first setup used agarose gels which were preemptively moved to a 28 ± 1 °C incubator. At 22–23 hpf, embryos were moved into custom agarose wells with the addition of 2–3 mL exposure media and the lid was closed and put back into the incubator. Zebrafish TC activity was recorded at ~24 hpf for 1 min using a charge-coupled device (CCD) camera (Zgenebio, Taipei, Taiwan) mounted to a dissecting microscope (Shenzhen Saike Digital Technology Development Co., Ltd., Shenzhen, China) within an incubator set to 28 ± 1 °C with a total magnification of 12×. Video was recorded at 1280 *×* 720 p at 60 fps in .mp4 format.

The second setup was meant for a large-scale tail coiling screening. In this setup, we used a 4K digital CCD Camera (Zgenebio, Taipei, Taiwan) mounted to a dissecting microscope (Shenzhen Saike Digital Technology Development Co., Ltd., Shenzhen, China) with a total magnification of 6× (Figure A1). Custom agarose wells were not used in this setup due to their size which limited the area for recording. The zebrafish used in this setup was also aged ~24 hpf. The video was recorded at 2560 *×* 1440 p at 30 fps in .mov format. Video recordings from both setups were converted to 30 fps, .avi format using VirtualDub2 software (http://virtualdub2.com/, accessed on 29 July 2021), due to ImageJ’s limitation in reading video format.

### 2.5. Data Collection Using ImageJ and Microsoft Excel 2016

ImageJ FIJI builds [19] (available online: https://https://imagej.net/Fiji/Downloads, accessed on 29 July 2021) and Microsoft Excel 2016 were used to collect and process the data. Several preinstalled tools and downloadable plugins are necessary for the data extraction process (standard operation protocol can be found in the Appendix A). The downloadable plugins used in this method are Hough Circle Transform [20], Highpass filter [21], BAR [22] and Read and Write Excel Plugins [23]. The data extraction process starts with automatic ROI selection that used the Hough Circle Transform plugin. Hough Circle Transformation method was modified from a previous study reported by Zhang et al., [16] in order to accommodate with the lower video quality. In the modified Hough Circle Transformation method, Gaussian blur was used to increase the area covered by zebrafish chorion, afterward, IsoData mode rather than Otsu was used in the thresholding step, which better suits the video with low contrast background for Hough Circle Transform. The modified method is also more suitable for data collection without determining the number of eggs at the start, which previously needs to be inputted manually during the data collection process. However, with the modified method by using the centroid map and threshold tool, it is possible to obtain all the eggs which are available in the image automatically (Figure 1B). These ROIs are saved in ImageJ “ROI Manager” and the “Results” window which will be used for the next steps. After the automatic ROI selection step, the video was further edited using Gaussian blur, unsharp mask and highpass filter to obtain a better signal-to-noise (SNR) ratio.

The resulting videos were subjected to StackDifference Plugin to generate a subtracted image stack which will show the difference between video frames. Next, Plot *Z*-axis Profile tool was used to obtain the plot on each saved ROI obtained from the automatic ROI selection process (Figure 1C, top panel), and BAR Plugin was used to extract the TC occurrence (Figure 1C, bottom panel) with a 30 frame or 0.5 s rest time for each TC occurrence which is a modified value from previous research [14]. The final results were exported to Excel (.xlsx) file using Read and Write Excel Plugin for the next processing step. Processes in ImageJ were compressed in one macro to provide a faster and mostly automatic (90%) process except for the initial steps which are the egg diameter selection and threshold setting. It took 2–3 min to run these processes for 1280 × 720 p video for each iteration using Intel i7-9700K processor, 32 GB of RAM, Windows 10 Pro desktop computer and ImageJ v1.53c.

During the early setup of the procedure, it might be necessary to rerun the BAR Plugin several times to obtain the optimal settings. If data correction is necessary, Plot *Z*-axis profile and BAR Plugin can be reconducted in succession to inspect the peaks and collecting the data with adjusted BAR configuration. Afterward, the data saved in Microsoft Excel should be manually replaced before the next step (TCMacro scripts are available in Appendix A).

Afterward, counting TC frequency, calculating average intensity, coil duration and the interval between tail coiling were carried out using Microsoft Excel 2016 software. This step was conducted by counting the number of maxima peaks which were calculated by BAR plugin using Microsoft Excel 2016 count function for each individual and dividing the result with the recording duration, average intensity by calculating the obtained intensity of each TC occurrence, while coil duration and the interval between coil are calculated using their own calculation method (Figure 1D). These processes are also automated using Excel Visual Basic for Applications (VBA) (TCMacro VBA scripts are available in Appendix A). A depth explanation of the method can be seen on the provided tutorial and standard operation protocol in Appendix A.

### 2.6. Statistical Analysis

Statistical analysis was carried out in GraphPad Prism 8 software (Graphpad Holdings, LCC, San Diego, CA, USA). The significance level was set at 0.05 (5%). Pearson correlation was used to calculate the correlation while, Wilcoxon test was used to calculate the difference between the proposed method and the manual calculation method as the data follows normal distribution assumption, presented as Mean *±* SD and Mean *±* SEM, respectively. The data were processed using Kruskal-Wallis due to not following the normal distribution assumption and presented as Mean *±* SD (**** *p* < 0.0001).

## 3. Results

### 3.1. Automatic Region of Interest (ROI) Selection Using Hough Circle Transform Plugin in Bright-Field Microscope Recording

Automatic ROI selection was performed using Hough Circle Transform Plugin in ImageJ FIJI Build. Our procedure employs a Gaussian blur tool to increase the edge area of each chorion in order to enhance the image for the thresholding process and Hough Circle Transform Plugin (Figure 2A,B). This step can do beneficial especially for those videos with low object-to-background contrast. Furthermore, we also tested our Automatic ROI selection capability for recording more eggs at once (Figure 2C). By using the optimized settings at 0.7 Hough Score Threshold, promising results were obtained for ROI selection with high precision and accuracy. On the contrary, other sub-optimized settings such as either lower Hough Score Threshold (0.5) or higher Hough Score Threshold (0.9) will lead to false positive or false negative errors for ROI selection (Figure 2C).

### 3.2. Automatic ROI Selection Using Hough Circle Transform Plugin in Using Dark-Field Microscopy Method

We tested our method further using a new recording setup that uses dissecting microscopy with darkfield effect settings. In contrast with the bright-field upright microscope (usually with 40× magnification for low magnification view), dissecting microscopy equipped with dark field effect can be used to record more embryos (around 100–150 embryos) at a 6–60× zoom magnification range with more flexibility. The result of implementing darkfield effect seems promising, as it is possible to record embryos with a high signal-to-noise ratio (Figure 3A).

When zebrafish embryos were recorded by the above microscope setting, we found the relative position of some embryos was altered due to high TC activity, and this movement will make TCMacro difficult to select their identifies (Figure 3B). In order to reduce the embryo movement during video recording, two major approaches were adapted by using either smaller space to restrict embryo movement or mounting medium to increase solution viscosity. Limiting the area using smaller agarose holders indeed can reduce embryonic movement. However, the area should be tightly packed with eggs, otherwise, there will still be room for egg movement (Figure 3C,D). In addition, the limited space significantly reduces the total number of zebrafish embryos that can be recorded. Methylcellulose, on the other hand, increases the solution viscosity, which reduces the embryo movement. High methylcellulose concentration (1%) will result in a problem where the eggs will float above the others due to high viscosity, creating a layered effect due to embryo overlapped (Figure 3E). Finally, optimized concentration methylcellulose at 0.5% concentration gives the best immobilization effect showing almost no sign of embryo position shifting before and after video recording (Figure 3F).

### 3.3. Tail Coiling Activity Data Processing in ImageJ

After automatic ROI selection, the videos were subjected to further segmentation process for data collection. The data collection was carried out using Plot *Z*-axis profile to create a time-intensity plot from subtracted image stack for each automatically selected ROI and BAR Plugin for peak detection. Figure 4A represents the raw plot generated using Plot *Z*-axis profile before peak extraction using BAR Plugin. The plot shows changes of mean intensity during 1 min of recording (3600 frames). Within the plot, several distinct peaks, background noise and minor peaks can be observed.

Afterward, BAR Plugin was used to collect the peaks, which represent TC occurrence. Minimal amplitude and peak distance were set as data collection thresholds. Minimal amplitude was set to differentiate between peaks caused by TC occurrence and background noise. Meanwhile, peak distance is a minimal distance between peaks for it to be counted as one. In this example, we set the minimal amplitude to 0.95 and peak distance to 10 (Figure 4B) and 30 (Figure 4C). The BAR Plugin will determine the maxima (red dot) and minima (blue dot) points of the plot according to the minimal amplitude and peak distance set. From our result, Figure 4B with a peak distance of 10 resulted in 7 peaks while Figure 4C with a peak distance of 30 resulted in 6 peaks. The difference can be observed on the 3rd to 4th peak. Due to the 4th peak is situated less than 30 frames from the 3rd peak, therefore it is counted as a peak in Figure 4B but not a peak in Figure 4C.

In the following validation procedure, we used a minimal amplitude of 0.6–0.8 and a peak distance of 30. Minimal amplitude was set to 0.6–0.8 from our trial using the available setup, while the peak distance was 30 frames or 0.5 s which is modified from previous research as a rest time for each TC occurrence [14]. Both minimal amplitude and peak distance values are flexible, as they can be modified according to the video recording.

### 3.4. Tail Coiling Data Processing Using Microsoft Excel Visual Basic for Application (Excel VBA)

After peak data processing, TC occurrence and intensity can be obtained from ImageJ outputs in the form of an excel file. Later, another two important endpoints as the interval between TC occurrence and duration of TC occurrence can be obtained by mathematical calculation. The workflow of this step is presented in Figure 5. The data processing step in Microsoft excel was divided into 3 macros, which are ProcessItvAndDur, ManualEditing and DataExtraction, in order. ProcessItvAndDur is used to extract the TC interval and duration with pre-set thresholds. These thresholds are divided into start and end thresholds to obtain TC duration. Meanwhile, the data that did not meet these threshold requirements are counted as TC intervals. From our experiments, we set the start threshold to 3 which means if the intensity rose by 3 points it will start counting as duration. The end threshold is set to the median intensity value of the whole recording, therefore if the intensity reached the median value, the macro will stop counting the value as duration, thus counting it as TC interval (Figure 6) (For note, several recordings might have 0 or 1 TC occurrence which will translate to infinite interval or incalculable interval, the result will be presented as empty for statistical calculation). These values, however, will be different according to the recording setup and furthermore, if the data has a lot of noise, sometimes it will interfere with the result. Therefore, we provided a ManualEditing macro for manual editing. Manual intervention is necessary for this step as the errors must be managed by changing the threshold as needed. This tool will rerun the interval and duration with a new start and end threshold set by users. The results presented in Figure 4 are an example of a case where the start threshold is too high, which might result in an undetected 2nd peak, thus we adjust the value of the start threshold to 1.5 during the calculation process in order to accommodate the peak.

After inspecting and editing all errors, the final macro, DataExtraction is used to create several sheets namely, TC, Intensity, Interval, Duration and Summary. The first four sheets are used to extract respective information to each sheet, meanwhile in the summary sheet, TC frequency and average coil intensity, interval and duration for each fish embryo will be calculated and presented, providing the final data collection summary as the sheet name implies. Finally, those curated data can be used to perform statistical analysis for comparing the potential differences.

### 3.5. Method Validation Using MS-222 and Caffeine

After method establishment, we performed validation with MS-222 to see whether good data consistency can be obtained between manual counting and our developed TCMacro methods. MS-222 exposure on zebrafish embryo showed significant TC activity inhibition (Figure 7A). The result showed a reduction of TC activity in a dose-dependent manner compared to tail coiling activity in control zebrafish. Zebrafish embryos exposed to 0.1 ppb of MS-222 had their TC frequency reduced to 95.9 ± 15.2% of control TC frequency, while zebrafish embryos exposed to 1 ppb and 1 ppm MS-222 had their TC frequency reduced to 53.1 ± 9.9% and 46.5 ± 7.7%, respectively. On the other hand, there are no significant difference observed in coiling intensity, coil duration of control (10.2 ± 3.5, 0.8 ± 0.3 s) to 0.1 ppb (10.4 ± 3.8, 0.9 ± 0.3 s), 1 ppb (10.0 ± 5.4, 0.8 ± 0.4 s) and 1 ppm (10.2 ± 5.0, 0.9 ± 0.4 s) of MS-222 exposure (Figure 7B–C). However, a significant difference was observed on TC interval at 0.1 ppb MS-222 exposure (Figure 7D). MS-222 exposure increased delay of tail coiling interval from 8.5 ± 5.2 s of control to 15.3 ± 8.2 s of 0.1 ppb, 20.8 ± 10.9 of 1 ppb and 17.9 ± 11.1 of 1 ppm. The time-lapse representation of this data can be seen in Figure 7E,G for control and Figure 7F,H for 1 ppm MS-222 exposed fish. Afterward, we also used the obtained data to compare the result with the manual counting method which is one of the most common methods in detecting tail coiling then we used Pearson correlation calculation and the Wilcoxon test to validate the result from our method with compared to the manual counting method. The result from our double validation on control (r = 0.9936, ns), MS-222 0.1 ppb (r = 0.9925, ns), 1 ppb (r = 0.9995, ns) and 1 ppm (r = 0.9996, ns) showed no significant difference between both methods (Figure 7I–L). Therefore, the result obtained from the automated TCMacro method is consistent with the manual counting method.

Furthermore, we also tested caffeine as a positive control on boosting zebrafish embryos tail coiling activity. The test result showed that there was an increase in of TC frequency on zebrafish embryos exposed to caffeine. Compared with control, TC activities elevation were observed to a level of 435.4 ± 98.51%, 325.3 ± 90.82% and 196.1 ± 46.34%, respectively, when embryos were exposed to either 150, 300 or 600 ppm caffeine (Figure 8A). On the contrary, coil intensity parameter has a declining trend, starting from control group (15.9 ± 11.6) to caffeine exposed groups at either 150 ppm (12.3 ± 4.9), 300 ppm (11.3 ± 4.7) or 600 ppm (6.5 ± 7.2). By statistical analysis, 600 ppm caffeine exposed group showed a significant reduction of coiling intensity compared to control (Figure 8B). Significant increase of coil duration was also observed at 150 ppm (1.0 ± 0.95 s) and 300 ppm (1.2 ± 0.8 s) caffeine exposed group when compared to control (0.6 ± 0.4 s). On the other hand, the tail coiling duration for 600 ppm caffeine group (0.7 ± 0.9 s) showed no difference with control (Figure 8C). For tail coil interval, we found all caffeine exposed groups showed a significant reduction of tail coiling interval, indicating zebrafish embryos display more tail coiling events. The control group had a tail coil interval of 8.1 ± 4.2 s, while those in caffeine exposed groups were significantly shorter as 3.7 ± 3.3 s for 150 ppm, 4.5 ± 4.7 s for 300 ppm and 4.2 ± 4.9 s for 600 ppm (Figure 8D). Finally, we also added time-lapse representation for peaks and images in Figure 8E,G for control, and Figure 8F,H for 300 ppm caffeine exposed fish embryos on supporting TCMacro able to detect elevating TC events in zebrafish embryos after exposing to psychoactive compound.

## 4. Discussion

The most important finding for this paper is that we provided a simple and cost-effective method to conduct TC measurement in zebrafish embryos with great flexibility for the first time. After setup and parameter optimization, our provided TCMacro offers scientists able to conduct TC measurement in an automated and high-throughput manner. First, our optimized Hough Circle Transform setting showed a decent result for the ROI selection, which can select embryos with very high precision compared to the previous method reported by Zhang et al. [18]. Afterward, we overcome the low noise-to-signal ratio video quality problem by employing dissecting microscopy with darkfield illumination. With the aid of the low magnification zoom function of dissecting microscope, it is possible to do a large-scale measurement (n number can reach more than 100 embryos) of TC events for zebrafish embryos from single video tapping. This high n number can significantly enhance the statistical power for data analysis. In this paper, we also provide a good solution to overcome the problem due to embryo position shifting that may interfere with the data processing. The methylcellulose mounting method was identified as a better solution with fewer caveats compared to agarose on preventing noise raised from embryo position shifting.

In order to validate our method, we used Tricaine methanesulfonate (MS-222) to inhibit zebrafish embryo movement. MS-222 is a common anesthetic used in zebrafish for surgical procedures and temporary immobilization. It is the only anesthesia to be approved by the US Food and Drug Administration (FDA) for use in aquatic animals [18]. MS-222 works by blocking sodium and potassium currents in nerve membranes, limiting action and muscle contraction of the target [24]. As a euthanasia agent for zebrafish, it worked by blocking gill ventilation which might result in death due to hypoxia [18]. However, immature gill in early life zebrafish proved to increase their resistance toward MS-222, and zebrafish embryos still can obtain sufficient oxygen from surrounding water via diffusion. MS-222 test showed a significant reduction of TC activity on zebrafish embryo, supporting TCMacro indeed able to detect TC activity reduction. Furthermore, we also tested TCMacro and manual counting methods by using Pearson correlation and Wilcoxon test as double validation. The result shows high correlation and no difference between both methods, suggesting the proposed TCMacro method indeed suitable for TC activity data collection with high accuracy.

We also tested the effect of caffeine on zebrafish embryos TC activity as a positive control. Caffeine is widely known as a psychoactive stimulant that enhances physical performance, however, it is also known to create dependence and causing sleeping problems [25,26]. It works as an antagonist to adenosine receptors which mainly promotes sleep [27]. From our tests, we observed TC activity elevation accompanying with caffeine administration up to 300 ppm. However, at a higher dose (600 ppm) of caffeine, the excitation stimulating effect for zebrafish embryo tail coiling was compromised. These results are to be expected as a previous study showed a similar biphasic effect on zebrafish larvae [28]. These results support the capability of our proposed TCMacro method to accurately collect zebrafish TC data.

Even though the result of our testing showed promise, however, some essential key points should be kept in mind on using TCMacro for TC measurement. First, it is worth noting the fine adjustments for embryo size, ROI collection, threshold setting and TC data extraction using BAR Plugin within the macro string provided are still considered necessary for difference recording instrument setup. All those parameters have to adjust for the first time since the microscope and CCD setting might be different for every laboratory. As mentioned previously, a manual editing process is sometimes necessary during data processing. Several possible conditions that might need manual editing are,
The incident when the duration counting does not stop which is caused due to inadequate end threshold value.The incident where duration counting stop before TC stops. This incident is caused due to end threshold value set at excessive value.The incident when duration counting did not start, which is happened due to the start threshold value is set too high.The incident when there is a lot of duration with small values. This problem is caused due to start threshold being set too low. This incident will result in a lot of “false durations”.

These problems are usually caused due to the fluctuating noise level of the recording. Therefore, it is recommended to use a good recording setup to reduce troubleshooting during data processing. Furthermore, using a camera with a wider field-of-view is recommended to accelerate the data collection process. The use of agarose wells in this study is to ensure no outside disturbance from machine vibration, which necessity might depend on the condition, hence it is possible to record without the agarose wells to record more eggs at once as presented in our large-scale recording setup which uses methylcellulose or agarose walls for reducing egg movement due to TC occurrence. In summary, compared to the previous published ImageJ-based method [16], our newly invented TCMacro method does not require an internet connection after the installation process and users can do some adjustments to the process according to their preference.

## 5. Conclusions

TCMacro, a simple and cost-effective Image J-based method proposed here opens a new avenue to conduct high-throughput TC data collection and extraction in zebrafish for the first time. Since the TCMacro script is based on the ImageJ platform, it provides the research community an alternative for zebrafish TC measurement with free-to-use, customizable and user-friendly advantages. The current version processes are semi-automatic (around 90%), where the manual operation is only at the start of each step. Thus, this method will increase the repertoire and adding more options for collecting TC data for research. Appendix A can be accessed for more information on the use of TCMacro which included an in-depth step-by-step guide for our tool and the macro package for ImageJ and Microsoft Excel 2016.

## Figures and Tables

**Figure 1 biomolecules-11-01133-f001:**
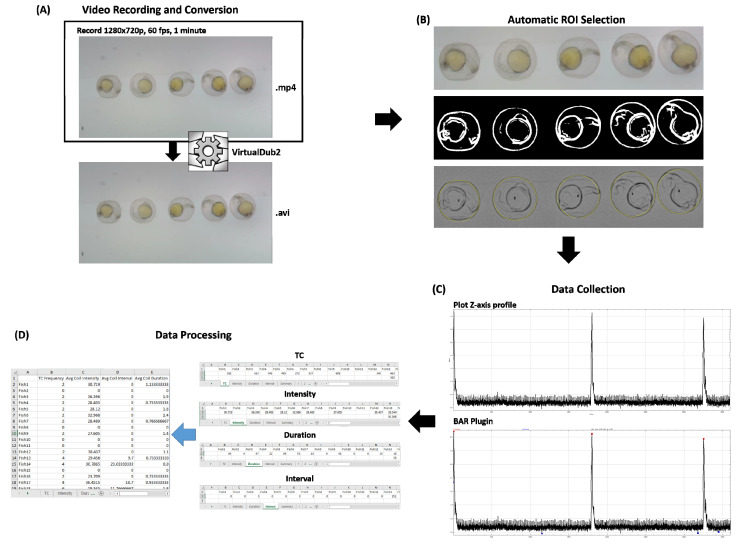
Plotline workflow of TCMacro for zebrafish embryo tail coiling measurement. The process starts with video recording (**A**), automatic ROI Selection (**B**), data collection using Plot *Z*-axis profile and BAR Plugin (**C**) and the collected data was saved as Excel file (.xlsx) for data processing in excel (**D**).

**Figure 2 biomolecules-11-01133-f002:**
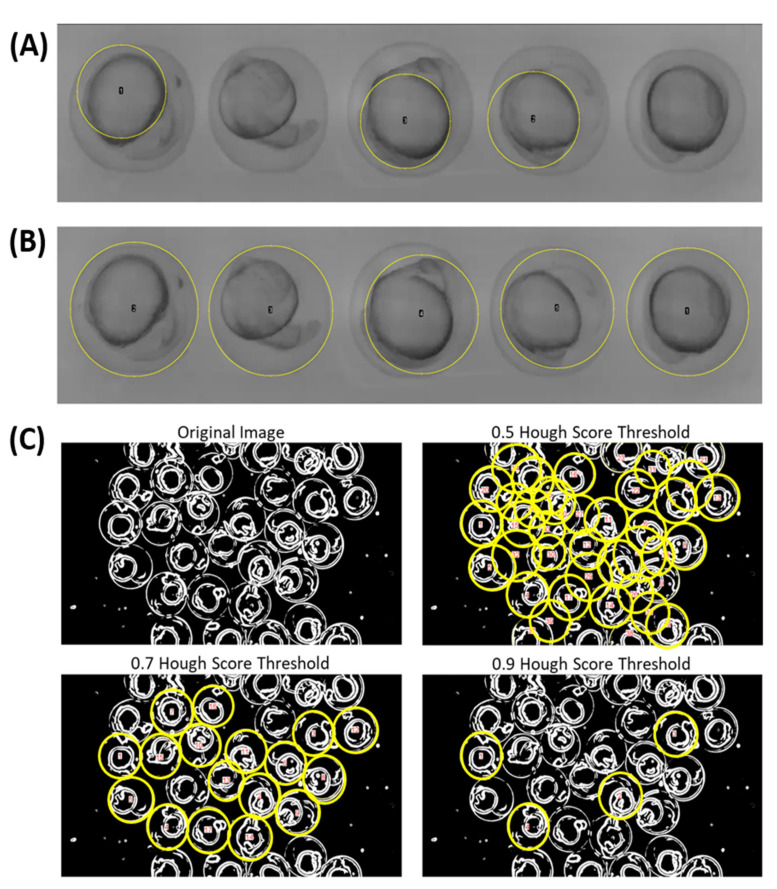
Optimization of Hough Circle Transformation for object circularity detection. Hough Circle Transform result according to (**A**) Zhang et al., 2019 method or (**B**) our proposed TCMacro method (Hough Score Threshold = 0.5). (**C**) Hough Score Threshold value fine-tuning to acquire the best automatic ROI selection result. Recorded at 12× magnification. Eggs be correctly selected were labeled with yellow circles.

**Figure 3 biomolecules-11-01133-f003:**
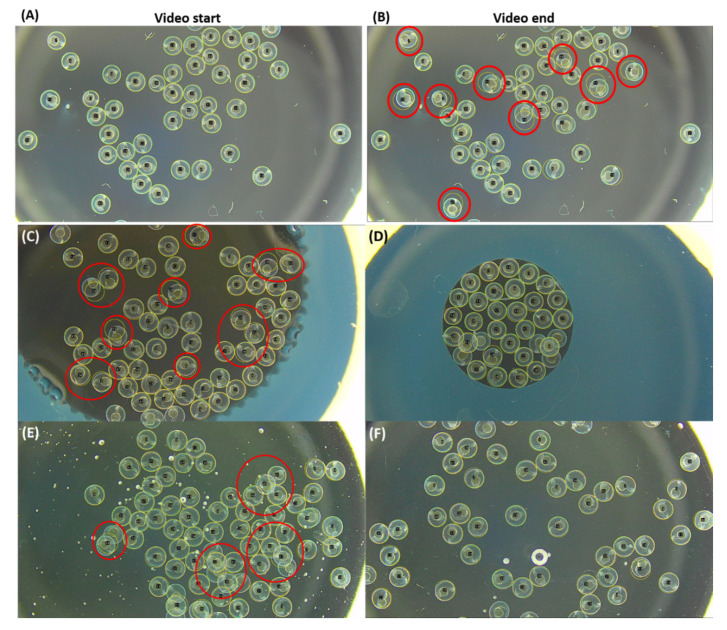
Comparison of different methods to reduce embryo position shifting during video recording for tail coiling measurement in zebrafish. (**A**) ROI selection in dissecting microscopy with darkfield setting. (**B**) Some embryos’ position was shifted due to embryo tail coiling (labeled by red circles). Two different sizes of agarose holders were tested to reduce the inter-embryo space with either (**C**) agarose outer wall (ø = 8.5 mm) or (**D**) agarose outer wall (ø = 4.4 mm). Two different concentrations of methylcellulose at either (**E**) 1% or (**F**) 0.5% were used to increase the mounting medium viscosity to prevent embryo position shifting. Egg movement caused by tail coiling occurrence was detected and marked with red circles. Images were captured at a low magnification at 6×.

**Figure 4 biomolecules-11-01133-f004:**
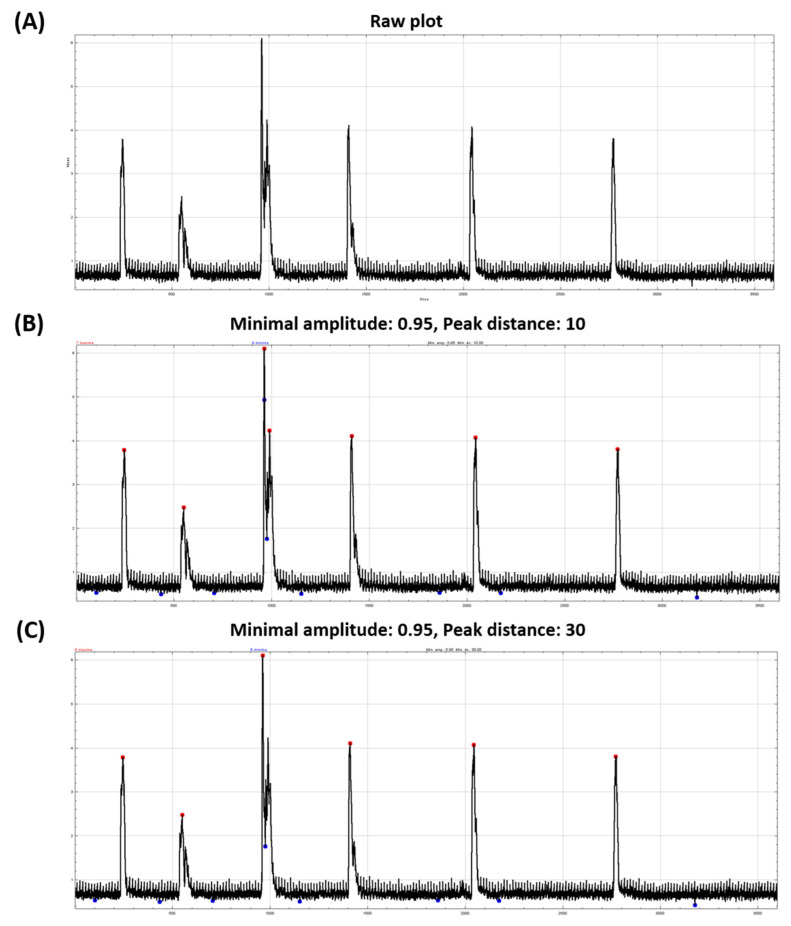
Tail coiling activity plot over time for zebrafish embryo tail coiling activity measurement. The *x*-axis represents time (frame), while the *y*-axis represents signal intensity. Raw tail coiling activity plot obtained from Plot *Z*-axis Profile tool (**A**) and peaks extracted at 0.95 minimal amplitude and different peak distance of 10 (**B**) and 30 (**C**), *x*-axis = video frames, *y*-axis = intensity.

**Figure 5 biomolecules-11-01133-f005:**
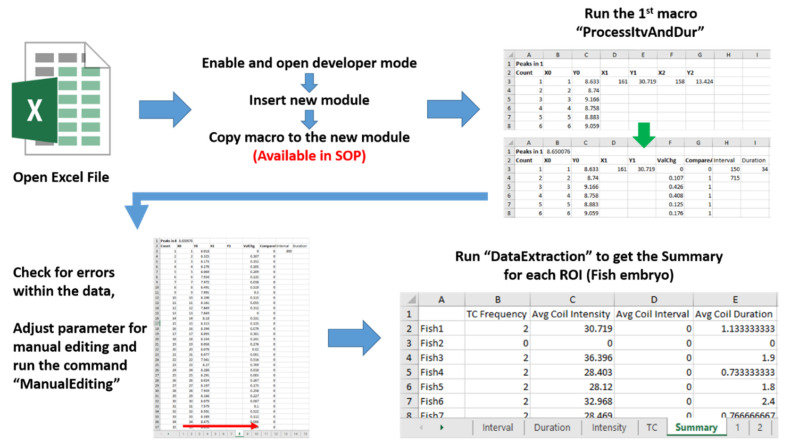
Workflow of tail coiling data collection automation step using Microsoft Excel VBA (Visual Basic for Applications). Three important macro scripts of ProcessitvAndDur, ManualEditing and DataExtraction were developed to boost data calculation automation.

**Figure 6 biomolecules-11-01133-f006:**
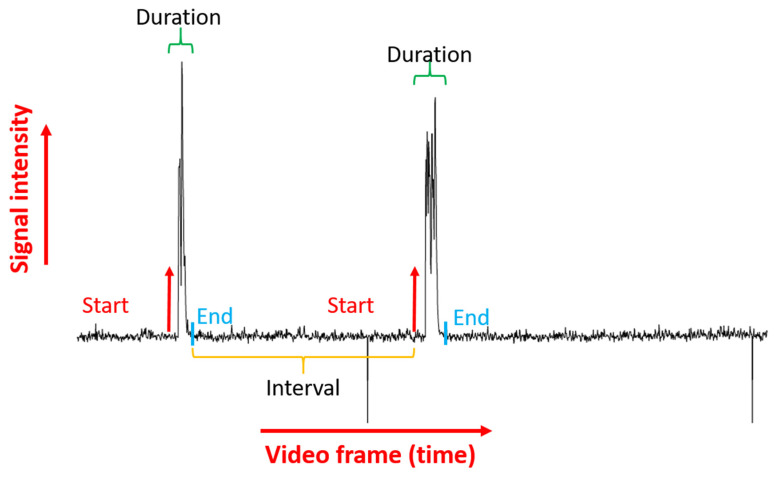
Start and end threshold set for tail coiling occurrence duration and the interval between tail coiling occurrence. Some important endpoints such as tail coiling interval and tail coiling duration can also be extracted from the raw signal intensity over time data.

**Figure 7 biomolecules-11-01133-f007:**
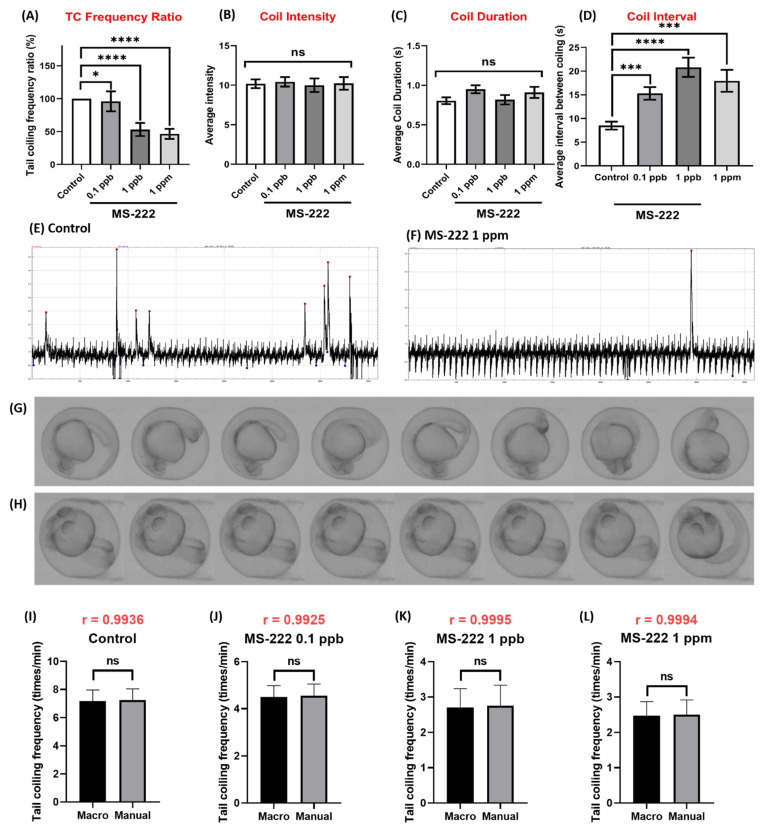
Comparison of tail coiling activity for zebrafish embryos at 24 hpf for control and MS-222 exposure. (**A**) Quantitative comparison of tail coiling frequency ratio, (**B**) average intensity, (**C**) coil duration and (**D**) coil interval of zebrafish embryos treated with different doses of MS-222. The representative tail coiling peak obtained from either (**E**) control or (**F**) 1 ppm MS-222 exposed zebrafish. Time-lapse comparison showing tail coiling movement in zebrafish embryo after receiving either (**G**) 0 or (**H**) 1 ppm MS-222. Statistical validation is carried out using Kruskal-Wallis test. Data are presented as Mean ± SEM, *n* = 40. (**I**–**L**) Statistic comparison of data collected by either TCMacro or manual counting for either (**I**) control, (**J**) MS-222 0.1 ppb, (**K**) MS-222 1 ppb or (**L**) MS-222 10 ppb treated embryos. The statistical difference is double validated using Pearson correlation (black color) and Wilcoxon test (red color). Data are presented as Mean ± SEM, *n* = 40 (* *p* < 0.05, *** *p* < 0.001, **** *p* < 0.0001, ns *p* > 0.05).

**Figure 8 biomolecules-11-01133-f008:**
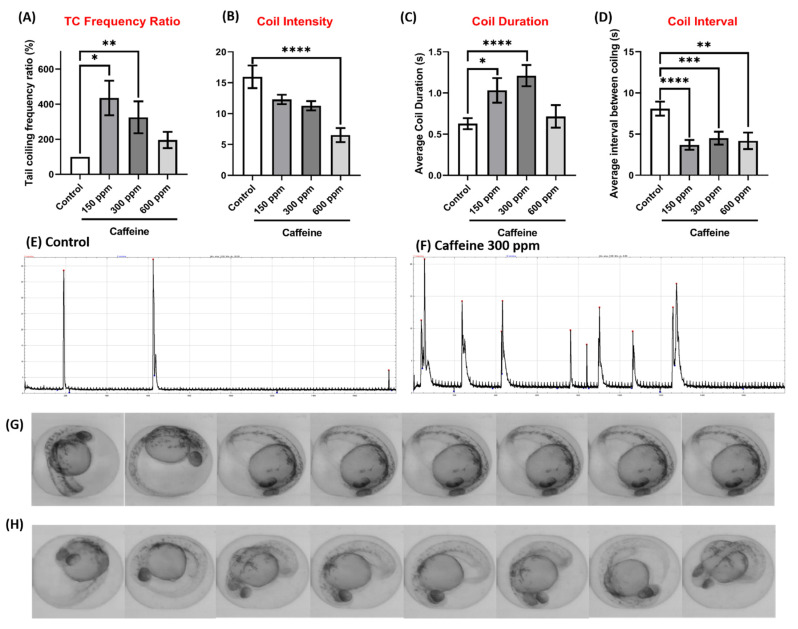
Comparison of tail coiling activity for zebrafish embryos at 24 hpf for control and caffeine exposure. (**A**) Quantitative comparison of tail coiling frequency ratio, (**B**) average intensity, (**C**) coil duration and (**D**) coil interval of zebrafish embryos treated with different doses of caffeine. The representative tail coiling peak obtained from either (**E**) control or (**F**) 300 ppm caffeine exposed zebrafish. Time-lapse comparison showing tail coiling movement in zebrafish embryo after receiving either (**G**) 0 or (**H**) 300 ppm caffeine. Statistical validation is carried out using Kruskal-Wallis test. Data are presented as Mean ± SEM, *n* = 40 (* *p* < 0.05, ** *p* < 0.01, *** *p* < 0.001, **** *p* < 0.0001, ns *p* > 0.05).

**Table 1 biomolecules-11-01133-t001:** Comparison of available methods for obtaining tail coiling activity in zebrafish.

Paper	Platform	Free Accessible?	Feature	Processing Speed	Endpoints	Internet Connection
Our method	TCMacro, ImageJ and Microsoft Excel 2016	Yes	Automatic ROI selection, detection via pixel changes	2–3 min for 1 min, 1280 × 720 p, 60 fps video	Tail coiling activity, coil duration, interval and coiling intensity	No, only for software update
Bakar et al., 2017 [10]	Manual	Yes	Manual detection	Not available	Tail coiling activity	No
Chen et al., 2012 [11]	Manual	Yes	Manual detection	Not available	Tail coiling activity	No
Wang et al., 2019 [12,13]	Manual	Yes	Manual detection	Not available	Tail coiling activity	No
Zindler et al., 2019 [6,7]	DanioScope	No	Automatic ROI selection, detection via pixel changes	Not available	Tail coiling activity and coil duration	No
González-Fraga et al., 2019 [14]	ZebraSTM, MATLAB^®^	Yes	Detection via pixel changes	Not available	Tail coiling activity	No
de Oliveira et al., 2021 [15]	DanioScope	No	Automatic ROI selection, detection via pixel changes	Not available	Tail coiling activity and coil duration	No
Zhang et al., 2021 [16]	EMAnalysis, ImageJ and Website	Yes	Automatic ROI selection, detection via pixel changes	30 s for 20 s, 30 fps video	Tail coiling activity, coil duration, interval, coiling intensity and differentiation between live and dead zebrafish eggs	Yes, for data processing
Ogungbemi et al., 2021 [17]	KNIME	No	Automatic ROI selection, detection via pixel changes	Not available	Tail coiling activity	No

## Data Availability

The data presented in this study are available on request from the corresponding author.

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
