# Peer review of "TCMacro: A Simple and Robust ImageJ-Based Method for Automated Measurement of Tail Coiling Activity in Zebrafish"

_biomolecules, 2021, doi:10.3390/biom11081133_

Round 1

Reviewer 1 Report

This is an interesting study where authors use simple and cost-effective technology to measure tail-coiling activity of zebrafish embryos. The authors have been able to provide a new method to analyze with several upgrades compared to previous reports. However, the paper contains some shortcomings in regard to experimental data which could be improved. The authors used tricaine which is recommended as a general anesthesia for aquatic organisms. Tricaine shows hypoactivity in tail coiling of zebrafish embryos which authors have shown as a negative control in the experiments. However, hyperactivity is also an important aspect of the spontaneous tail coiling assessment in zebrafish behavioral studies. 

This new assay could be more helpful to the zebrafish research community, if the authors can introduce a positive control.

For example, Evdokia Menelaou et al., described the effects of KCl in zebrafish tail coiling; Embryonic motor activity and implications for regulating motoneuron axonal pathfinding in zebrafish. Eur J Neurosci. 2008 Sep;28(6):1080-96.  doi: 10.1111/j.1460-9568.2008.06418.x. 

KCl (potassium chloride) is known to cause depolarization in the spinal motor neurons and results in hyperactivity in zebrafish tail coiling.

Overall, the manuscript is well written and interesting.

Minor: There is repeatation of ref. 7 and 15.

Typo in line 86: The 86 dish wash covered... > was

Author Response

Comments and Suggestions for Authors

This is an interesting study where authors use simple and cost-effective technology to measure tail-coiling activity of zebrafish embryos. The authors have been able to provide a new method to analyze with several upgrades compared to previous reports. However, the paper contains some shortcomings in regard to experimental data which could be improved. The authors used tricaine which is recommended as a general anesthesia for aquatic organisms. Tricaine shows hypoactivity in tail coiling of zebrafish embryos which authors have shown as a negative control in the experiments. However, hyperactivity is also an important aspect of the spontaneous tail coiling assessment in zebrafish behavioral studies. This new assay could be more helpful to the zebrafish research community, if the authors can introduce a positive control.

For example, Evdokia Menelaou et al., described the effects of KCl in zebrafish tail coiling; Embryonic motor activity and implications for regulating motoneuron axonal pathfinding in zebrafish. Eur J Neurosci. 2008 Sep;28(6):1080-96.  doi: 10.1111/j.1460-9568.2008.06418.x. 

KCl (potassium chloride) is known to cause depolarization in the spinal motor neurons and results in hyperactivity in zebrafish tail coiling.

Overall, the manuscript is well written and interesting.

The authors appreciated the reviewer for the comments and suggestions, as suggested; a positive control had added in this revised manuscript. The authors used caffeine as a positive control in the manuscript as it has been tested previously as a candidate for locomotor activity positive control in zebrafish embryos (Figure 8). The authors also tested KCl in zebrafish as suggested, however, we were not able to obtain locomotor activity enhancement as in the ublication given by the reviewer, which is possibly due to the mutant zebrafish used by the study, which lacks normal embryonic motor output upon dechorination. Therefore, in this revised manuscript, we only updated caffeine data as positive control for boosting tail coiling activity.

Minor: There is repeatation of ref. 7 and 15.

Thank you for the reminder, the authors have edited the references to avoid repeated references.

Typo in line 86: The 86 dish wash covered... > was

Thank you for the comment, the authors have edited the manuscript for the typo.

Reviewer 2 Report

The authors present a new and improved script to measure zebrafish embryo tail-coiling function in large-scale studies. While several tools already exist for this, the fact that it is completely FIJI and excel based (and thus  free) is a big advantage over some of the other tools. The authors also extract more information that can be easily obtained by manual TC analysis. The rationale of the script, options for manual fine-tuning, and validation are well described. It is also great to see that the outcome of TCMacro and manual analysis virtually the same. Overall, it appears that TCMacro is a nice addition to the field, and a great tool. Especially for researchers using TC as a readout in large-scale screening assays, for whom the fine-tuning might be a big advantage as the script can be tailored to their video setup. In addition, the authors also present some useful tweaks to the context in which the embryos are filmed for optional outcome.

I do have some comments regarding the manuscript, macro and validation studies that should be addressed before publication. I expect that the authors can easily resolve most (if not all) of them.

General comments:

  • Use of Tricaine, in the abstract referred to as neurotoxic. Tricaine is often used as sedative. Is the reduce TC due to neurotoxicity, or just sedation? I would guess sedation would already be sufficient to validate the TCMacro.
  • The authors present several options for manual editing or adjustments as positive features of their TCMacro. The reviewer is wondering if commercial tools like DanioScope also suffer from the same limitations as TCMacro that need the manual adjustments, or that the need for manual adjustments results from (minor) flaws in TCMacro?
  • Validation is only done with a sedative. It would be great if the authors could show that TCMacro also works when compounds that increase TC activity. I would be very much in favour of adding these data to the validation studies.

Methods:

  • Chemical Preparation and Exposure. Tricaine is stored at 4oC, but all data recommends freezing as it gets old very quickly. How can the authors ensure reproducibility of their experiments?
  • Video recording and conversion: why do the authors switch from 28 to 26oC in the middle of the procedure?
  • Recording setup for large-scale tail coiling screening & Video recording and conversion are partially redundant.
  • Lines 162-165: should be in results or discussion section, not methods.

Results:

  • Tail coiling Data Processing using Microsoft Excel Visual Basic for Application (Excel VBA): line 270: threshold starts to count upon 3-point increase. However, peak 2 in figure 2 does not reach the value of 3. What do the authors mean by 3 points increase? Or is this a limitation in the threshold analysis?
  • Positive about the previous comment is the implement option for manual editing.
  • Method validation using MS-222: please also present the MS-222 dose in mg/ml to allow comparison with doses used in sedation and euthanasia. Please also provide some perspective on the dose-range in the discussion.
  • Figure 7D. distribution of values from MS222 treated larvae does not seem to match with 7A. There should be a relation between frequency and interval. Why are there values of 0 (or close to 0) in the MS222 treated samples? This infers constant TC activity, which probably does not match with reality. Issue with the script?

Discussion:

  • MS222 doses, please provide some perspective in relation to doses for sedation and euthanasia. What is there any insight in potentially reduced MS222 update due to the fact these embryos are still in their chorion, and normally it is used on free swimming larvae?

Small issues:

  • Line 41, 44: embryo should be plural.
  • Line 49: use the term zebrafish larvae, instead of more mature zebrafish. 5-6 dpf is not mature at all.
  • Line 64: typo, we were able
  • Line 90-96: this section switches between past and present tense.
  • Line 101-102: rephrase Zebrafish eggs aged at 24 hpf is used in this step.
  • Figure 2C ROIs lack contrast, difficult to see the false-positives. Maybe the authors could emphasize the ROIs with thicker lines?
  • while in general good, there are quite some typos, extra spaces and missing words that I did not indicate here. Careful revision in this respect is recommended.

Author Response

Comments and Suggestions for Authors

The authors present a new and improved script to measure zebrafish embryo tail-coiling function in large-scale studies. While several tools already exist for this, the fact that it is completely FIJI and excel based (and thus free) is a big advantage over some of the other tools. The authors also extract more information that can be easily obtained by manual TC analysis. The rationale of the script, options for manual fine-tuning, and validation are well described. It is also great to see that the outcome of TCMacro and manual analysis virtually the same. Overall, it appears that TCMacro is a nice addition to the field, and a great tool. Especially for researchers using TC as a readout in large-scale screening assays, for whom the fine-tuning might be a big advantage as the script can be tailored to their video setup. In addition, the authors also present some useful tweaks to the context in which the embryos are filmed for optional outcome.

I do have some comments regarding the manuscript, macro and validation studies that should be addressed before publication. I expect that the authors can easily resolve most (if not all) of them.

General comments:

  • Use of Tricaine, in the abstract referred to as neurotoxic. Tricaine is often used as sedative. Is the reduce TC due to neurotoxicity, or just sedation? I would guess sedation would already be sufficient to validate the TCMacro.

Thank you for the comment. It is true that tricaine is often used as a sedative. However, the experimental design that the authors used in the current study was exposing the zebrafish embryos to tricaine at 6 hpf until 24 hpf. Therefore, the authors addressed it as a developmental neurotoxic material rather than a sedative.

  • The authors present several options for manual editing or adjustments as positive features of their TCMacro. The reviewer is wondering if commercial tools like DanioScope also suffer from the same limitations as TCMacro that need the manual adjustments, or that the need for manual adjustments results from (minor) flaws in TCMacro?

The authors appreciated the comments. Currently, the authors have no way to access DanioScope or other commercial tools to compare with our proposed TCMacro. However, from several papers that used DanioScope for collecting tail coiling data, there seem to be no manual adjustments necessary in DanioScope. The manual editing step on our method, as mentioned by the reviewer, is the result of flaws in TCMacro, which might be improved in the future from a hardware or software perspective.

  • Validation is only done with a sedative. It would be great if the authors could show that TCMacro also works when compounds that increase TC activity. I would be very much in favour of adding these data to the validation studies.

Thank you for the suggestion. As the reviewer suggested, the authors had added caffeine as a compound that increases TC activity. This compound was used as it has been previously demonstrated to increase TC activity. However, the authors also found out the biphasic effect of the compound as reported in the revised manuscript (Figure 8).

Methods:

  • Chemical Preparation and Exposure. Tricaine is stored at 4oC, but all data recommends freezing as it gets old very quickly. How can the authors ensure reproducibility of their experiments?

The authors appreciated the questions. The tricaine was stored at 4 ℃ for a maximum of one week of the experiment before preparing a new batch of stock solution, therefore, the authors were able to ensure the freshness of our tricaine solution that was used in the current experiment.

  • Video recording and conversion: why do the authors switch from 28 to 26oC in the middle of the procedure?

Thank you for the question. The correct temperature is 28 ℃ for the zebrafish egg incubation and for the recording. The authors made a mistake in typing the temperature as the adult zebrafish used for breeding were kept at 26 ℃.

  • Recording setup for large-scale tail coiling screening & Video recording and conversion are partially redundant.

The authors appreciated the suggestion. The authors had tried to summarize the method into one section to reduce the redundancy. However, some parts might still be redundant due to the similar setup that was used for both methods.

  • Lines 162-165: should be in results or discussion section, not methods.

Thank you for the comment, the authors had adjusted the sentences to the discussion section as suggested by the reviewer.

Results:

  • Tail coiling Data Processing using Microsoft Excel Visual Basic for Application (Excel VBA): line 270: threshold starts to count upon 3-point increase. However, peak 2 in figure 2 does not reach the value of 3. What do the authors mean by 3 points increase? Or is this a limitation in the threshold analysis?

Thank you for your question, the authors would first like to assume the reviewer meant for peak 2 on the 2nd graph in figure 4 as there is no graph with a peak in figure 2. As we mentioned in the next lines, the start threshold at 3, was meant for Figure 6 and most of our data. Furthermore, this condition is one of the reasons why the authors added the ManualEditing macro as the strength of coiling and recording setup have an effect on the intensity displayed in ImageJ. The authors had tried to elaborate more on this matter in the edited version.

  • Positive about the previous comment is the implement option for manual editing.

The authors appreciated the comment. It is true that this is why the authors implemented the manual editing macro for validating several conditions as the authors mentioned in the previous comment and discussion part.

  • Method validation using MS-222: please also present the MS-222 dose in mg/ml to allow comparison with doses used in sedation and euthanasia. Please also provide some perspective on the dose-range in the discussion.

Thank you for the comment, the authors had added the concentration of MS-222 in the previous study for sedation/developmental neurotoxic substance in the method part as it will be easier to directly compare with the concentration that the authors used. The concentration of MS-222 in previous study was 0.01 mg/mL, 0.05 mg/mL, and 0.1 mg/mL which are 100x timer higher compared to the concentration that we used (1x10-7 mg/mL, 1x10-6 mg/mL and 1x10-3 mg/mL). In the case of euthanasia, MS-222 is not suitable for euthanizing zebrafish embryos as according to a previous study, 1 mg/mL of the compound did not induce euthanasia.

  • Figure 7D. distribution of values from MS222 treated larvae does not seem to match with 7A. There should be a relation between frequency and interval. Why are there values of 0 (or close to 0) in the MS222 treated samples? This infers constant TC activity, which probably does not match with reality. Issue with the script?

The authors appreciated the questions. Previously, the authors did not take into account the 0 values. The 0 values here did not mean constant TC activity, rather it showed the interval between TC cannot be counted, therefore closer to infinity. However, the value “infinity” might be unsuitable to use in statistical tests. Due to this problem, the authors decided to keep the data empty for those with infinity. The authors had also considered having the start and end of the video as imaginary peaks to make the calculation easier, however, it will result in inaccurate interval calculation, the interval value would be close to:

Average interval = recording duration / (number of peaks +1) – average duration

Hence, the authors decided to keep the calculation and showing those with only 1 TC occurrence of no TC occurrence as “Nan” or empty in the statistical calculation. The authors had added a note for this matter in the manuscript.

Discussion:

  • MS222 doses, please provide some perspective in relation to doses for sedation and euthanasia. What is there any insight in potentially reduced MS222 update due to the fact these embryos are still in their chorion, and normally it is used on free swimming larvae?

Thank you for the question. As requested previously, the authors had added some perspective in relation to MS-222 doses for both sedation and euthanasia in the method part of the manuscript. Therefore, in order to reduce redundancy, the authors only mentioned a little bit about the sedation/euthanasia concentration in the discussion part. Secondly, a study has covered the role of chorion and perivitelline space (PVS) in zebrafish embryos as a defensive mechanism from external effects that slow down MS-222 penetration into the embryos.  

Small issues:

  • Line 41, 44: embryo should be plural.

Thank you for the comment. The authors have adjusted the manuscript according to the reviewer’s comment.

  • Line 49: use the term zebrafish larvae, instead of more mature zebrafish. 5-6 dpf is not mature at all.

Thank you for the comment. The authors have adjusted the term according to the reviewer’s suggestion.

  • Line 64: typo, we were able

Thank you for the comment. The authors have addressed the grammatical error as the reviewer suggested.

  • Line 90-96: this section switches between past and present tense.

Thank you for the comment. The authors have addressed the grammatical error as the reviewer suggested.

  • Line 101-102: rephrase Zebrafish eggs aged at 24 hpf is used in this step.

Thank you for the comment. The authors have changed “Zebrafish eggs aged at 24 hpf is used in this step” to “24 hpf zebrafish embryos were used in this step” as the reviewer’s suggestion.

  • Figure 2C ROIs lack contrast, difficult to see the false-positives. Maybe the authors could emphasize the ROIs with thicker lines?

The authors thanked the reviewer for the suggestion. The authors have tried their best to emphasize the ROI with thicker lines in this edited version. However, as we mark it manually by ourselves, as the thickness cannot be increased in ImageJ, some marking might not be perfect, therefore, the authors hope that the reviewers can understand this condition.

  • while in general good, there are quite some typos, extra spaces and missing words that I did not indicate here. Careful revision in this respect is recommended.

The authors appreciated the suggestion. Therefore, the authors had tried their best to edit the manuscript as the reviewer suggested.

Round 2

Reviewer 2 Report

Dear editor and authors,

 I am please with the answers and improvements. Particularly happy to see the caffeine treatment (figure 8) as a nice addition to show that TCmacro can reliably detect changes in both directions. I think that this work can be accepted for publication. 

Best wishes